# Optimized Dental Implant Fixture Design for the Desirable Stress Distribution in the Surrounding Bone Region: A Biomechanical Analysis

**DOI:** 10.3390/ma12172749

**Published:** 2019-08-27

**Authors:** Won Hyeon Kim, Jae-Chang Lee, Dohyung Lim, Young-Ku Heo, Eun-Sung Song, Young-Jun Lim, Bongju Kim

**Affiliations:** 1Clinical Translational Research Center for Dental Science, Seoul National University Dental Hospital, Seoul 03080, Korea; 2Department of Mechanical Engineering, Sejong University, Seoul 05006, Korea; 3Bio-based Chemistry Research Center, Korea Research Institute of Chemical Technology, Ulsan 44429, Korea; 4Global Academy of Osseointegration, Seoul 03080, Korea; 5Department of Prosthodontics and Dental Research Institute, School of Dentistry, Seoul National University, Seoul 03080, Korea

**Keywords:** implant design, dental implants, biomechanics, finite element analysis, primary stability, dental nerve

## Abstract

The initial stability of a dental implant is known to be an indicator of osseointegration at immediate loading upon insertion. Implant designs have a fundamental role in the initial stability. Although new designs with advanced surface technology have been suggested for the initial stability of implant systems, verification is not simple because of various assessment factors. Our study focused on comparing the initial stability between two different implant systems via design aspects. A simulated model corresponding to the first molar derived from the mandibular bone was constructed. Biomechanical characteristics between the two models were compared by finite element analysis (FEA). Mechanical testing was also performed to derive the maximum loads for the two implant systems. CMI IS-III active (IS-III) had a more desirable stress distribution than CMI IS-II active (IS-II) in the surrounding bone region. Moreover, IS-III decreased the stress transfer to the nerve under the axial loading direction more than IS-II. Changes of implant design did not affect the maximum load. Our analyses suggest that the optimized design (IS-III), which has a bigger bone volume without loss of initial fixation, may minimize the bone damage during fixture insertion and we expect greater effectiveness in older patients.

## 1. Introduction

In general, natural teeth are lost or partially damaged for reasons such as periodontal disease, dental caries, untreatable fractures, injuries, and orthodontic problems [1,2,3]. The dental implant systems have been mainly used to restore chewing motion during the past decades [4,5,6].

In 1969, the process of bone union through dental implants was first reported by Branemark et al. [7]. The initial treatment method required a 3–6 month unloaded period for successful bone union [8]. However, operation methods that require an unloaded period have problems such as several surgeries and a long treatment period. In order to solve these problems, early and immediate loading treatments have been developed [9]. Previous clinical studies for early and immediate loading have shown similar success rates to traditional treatments [10,11]. However, there is still no standard protocol for the loading range and stability of implants in early and immediate loading treatments. Several studies have reported the optimal effect of bone union [12,13,14].

In particular, the implant stability is considered to be an important factor for the healing process, bone union, and the ultimate success of the implant [15,16]. In the dental implant area, the implant stability is classified into treatment period, primary stability, and secondary stability [16]. The primary stability is defined by the initial fixation force determined by mechanical characteristics immediately after implant placement [17]. However, the secondary stability is defined as the formation of bone by a biological response at the bone–implant interface [17]. The overall stability consists of a combination of the primary and secondary stability, and if the initial primary stability is not sufficient, implant fixation failure may occur by the stability dip phenomenon [18]. Therefore, various biomechanical factors of the implant should be considered for effective bone union at the bone–implant interface.

In immediate loading, the implant design is an important factor in determining the stress distribution and primary stability during osteointegration [19,20,21]. The previous literature has reported consideration of factors such as the pitch, width, depth, shape of the thread, and crestal module with respect to the implant designs and osseointegration. The shape of the thread is generally composed of a buttress, reverse buttress, square shape, V-shape, etc., and the square form has been reported in the previous literature to be the optimal shape to provide a desired stress distribution in the bone [22,23]. In terms of the pitch of the thread, the optimal pitch size varies according to various thread shapes, and Lan et al. recommended that triangular- and trapezoidal-thread shapes are the optimal pitch sizes of 1.2 and 1.6 mm, respectively [24]. Kong et al. reported that the desired stress distribution was induced in the pitch size of a 0.8 mm for the V-shape [25]. Another study revealed that pitch size of 0.6 mm induces more crestal bone reduction than that of 0.5 mm [26]. According to the previous literature, the optimal thread width and depth are 0.18 to 0.3 mm and 0.35 to 0.5 mm, respectively [27]. Also, the stress concentration of the bone in the depth was found to be a sensitive factor more than the width [27]. 

As mentioned above, although the optimal size and shape were derived for each factor, a dental implant system consists of a combination of various factors, such as the shape, pitch, depth, and width size of the thread, the platform designs, and fixture type, etc. Hence, the optimal pitch size, fixture type, etc. can be changed. Therefore, comparative analysis should be performed considering overall design factors. 

We considered two different dental implant systems for optimized design. Firstly, IS-II implant system was designed to improve the initial fixation by including macro threads of the coronal area, a conical implant abutment seal, a deep buttress thread with 0.8 mm pitch, and a cutting edge with reference to the previous literature, as shown in Figure 1a. Additionally, an S-shape for greater formation of junctional epithelium and connective tissues was designed on the cervical part of the implant, as shown in Figure 1a.

In a study on comparative clinical trials of two different implants by Ryu et al., the equivalence of the IS-II and SLActive Bone level implant (Institut Straumann AG, Basel, Switzerland) were revealed through the results of determining the bone loss and implant stability quotient (ISQ) [15]. The initial fixation was not statistically significant in the ISQ, but the IS-II had a lower tendency [15]. To provide improvements, IS-III was newly developed. The deep buttress thread with a 0.9 mm pitch and the wide cutting edge were designed to provide greater bone volume and higher initial fixation, as shown in Figure 1b. In previous literature, it was identified that the stress concentration was highest in the cervical area of the fixture which was in contact with the cortical bone [26,27]. IS-II with an S-shape on the cervical area also has a thin section, so there is a high risk of stress concentration. To compensate for this disadvantage, the cervical design of IS-III was changed to a bevel shape to maintain the thickness of the cervical area, as shown in Figure 1b.

Our study was focused on comparing the biomechanical characteristics between IS-II and IS-III with respect to the design change using a universal testing machine and finite element analysis (FEA). The hypotheses of this study are that (1) IS-III has a more desirable stress distribution than IS-II and that (2) IS-III with a wide cutting edge causes the decrease of stress transfer to the nerve.

## 2. Materials and Methods 

### 2.1. The Design of the Implant System and Specimen Preparation

Two different fixture designs in terms of the collar, thread pitch, and cutting edge were selected for this study. IS-II (Neobiotec Inc., Seoul, Korea) presents an S-shaped collar design, and IS-III (Neobiotec Inc., Seoul, Korea) presents a micro-grooved design in the collar area and a wider cutting edge than the IS-II. The dimensions of the two different fixtures were identical at 4.5 mm in diameter and 10 mm in length. The fixture thread pitch was 0.8 mm in IS-II and 0.9 mm in IS-III, as shown in Figure 2c,d. Abutments and abutment screws of the same design were applied in both fixture models. The abutment has a length of 8 mm, as shown in Figure 2a, while the abutment screw has the screw pitch 0.4 mm, body diameter 1.95 mm, head diameter 2.3 mm, and length 8.8 mm, as shown in Figure 2b.

### 2.2. Finite Element Analysis (FEA)

In this study, two different fixture models and the remaining parts with the same dimension were used, as shown in Figure 2. To compare the stress distribution and primary stability within the bones for the two different fixtures, we utilized a 3D mandibular FEA model used in previous studies [28]. The cancellous and cortical bones was reconstructed in the same way as in the previous study [28,29]. According to the previous literature, occlusal force is mostly generated in the molar tooth [30]. Hence, the first molar tooth was extracted to reproduce a single surgical FEA model, as shown in Figure 3. The crown model was constructed in the same way in the previous literature [28]. A dental nerve of 2 mm in diameter was constructed 1 mm below the apex of the fixture in order to measure the deformation and stress distribution and of the nerve in the two models with respect to design changes in the apex cutting edge [31].

Software (ABAQUS CAE2016, Dassault systems, Vélizy-Villacoublay, Yvelines, France) was utilized to build the assembled model for each part, including the crown, implant systems, bone models, and nerve, as shown in Figure 3a,b. The material properties of the FEA models were used by referring to the previous studies, as shown in Table 1 [32,33,34,35,36,37].

A mesh of the surgical FEA model was formed using hypermesh software (Altair Hypermesh v19.0, Altair Engineering, Troy, MI, USA). Tribst et al. performed a mesh convergence study for a dental implant system and revealed optimal mesh size as 0.3 mm [38]. For that reason, the mesh size of the implant system and nerve has 0.15 mm as the maximal value. The mesh size in the implant-bone interface was set to 0.15 mm. The information of mesh size, elements, and nodes used in the present study are shown in Table 2.

The boundary conditions applied to the FEA models were completely constrained at all directions so that neither side of the cortical or cancellous bone was rotated or moved in any direction, as shown in Figure 4b. The interfaces of bone and implant, abutment and cement, cement and crown, and cortical bone and cancellous bone were considered as the “Tie contact” condition. This condition indicates that the two parts were fully combined so that neither separation nor sliding in the contact surface of the two parts occur [35]. The contact surfaces between implant systems were considered in a sliding state with friction at the interface. A friction coefficient of 0.5 was used [35]. The reason for the application of friction at the implant systems was that a micromotion between the implant surfaces might occur under a chewing loading [35]. In the abutment screw, compression force occurring axially by a preload of 200 N was used in order to tighten the abutment and fixture, as shown in Figure 4a [35]. According to the previous literature, major remodeling of the bone–implant surface was affected by oblique loading [39]. Additionally, the physiological ranges of load were reported as 100 and 30 N for the axial and the horizontal directions, respectively [40]. Thus, reflecting the force generated at various directions in the oral environment, loads of 100 N axially [27], 100 N at 15° [41], and 30 N at 45° [27] were applied for the first molar, as shown in Figure 4b.

The maximum equivalent stress (Max EQS) at the surface of the bone surrounding implants was measured [42]. A lower value of the Max EQS indicates that the bone around the implant is at lower risk of bone failure, and a higher success rate [42]. The peak von Mises stress (PVMS) values of the implant systems and cement were measured by comparing the two different fixture designs. Additionally, the bone volumes were calculated at the thread (pitch to pitch) and the apex region (cutting edge) of each fixture.

### 2.3. Mechanical Testing

A material testing machine (E3000, Instron^®^, Norwood, MA, USA) was used for static shear-compression testing through monotonic loading, as shown in Figure 5. Generally, all mechanical testing of the dental implant systems was performed according to the ISO 14801:2016 standard with a cylinder applicator that allowed free motion of a hemispherical loading member when loaded, as shown in Figure 5. Five specimens of each design implant system were used to test for static shear-compression load to compare the failure strengths of the two different systems. For specimen fixation, a central longitudinal axis (Line D–E) of the implant abutment was chosen at a 30° angle to the loading direction (Line A–B) of the cylinder applicator, as shown in Figure 5. The distance between the intersection of line D–E and line A–B and the fixation region of the fixture was set to 11 mm. The distal portion of the fixture at 3 mm away from the platform was fixed using a resin cement, assuming marginal bone loss after surgery, as reported [43].

Static shear-compression testing was performed with axial load at a velocity of 1 mm/min until the failure of the implant systems, while deriving a force–displacement curve [44]. Failure in the static test was defined as having a fracture in any part among the implant systems or a sudden force drop observed through the force–displacement curve [45,46]. 

### 2.4. Statistical Analysis

The Max EQS values of each node in the interface of bone and fixture are indicated as mean ± standard error of the mean (SEM), and independent sample t-testing was performed to compare between the two models. Additionally, the Max EQS value and the minimum principal strain of each node on the nerve outer surface were derived to compare deformations according to the fixture design. To compare the structural stability of the dental implant systems, the results obtained through mechanical testing are indicated as mean ± standard deviation (SD) and were compared using the independent sample t-testing. The significance level was set at a *p*-value of < 0.05. SigmaPlot (Systat Software Inc., San Jose, CA, USA) was used to perform the statistical analysis. 

## 3. Results

In the cortical and cancellous bones, the mean values of the Max EQS for various loading conditions are shown in Figure 6a,b.

The stress distribution of the cortical bone was statistically significantly higher in IS-II than IS-III under all loading directions (*p* < 0.001), as shown in Table 3. On the other hand, the stress distribution of the cancellous bone was statistically significantly lower in IS-II than IS-III under all loading directions (*p* < 0.001), as shown in Table 3. To evaluate the stress transfer to the nerve, the values of the mean Max EQS and the minimum principal strain of the nerve surrounding the cancellous bone after various loading conditions were measured, as shown in Figure 6c,d. In the axial loading direction, the stress distribution of the nerve was statistically significantly lower in the IS-III than IS-II (*p* < 0.001). However, there was no significant difference between the two models in both the 15° and 45° loading directions (*p* = 0.168, *p* = 0.775, respectively), as shown in Table 3. The minimum principal strain of the nerve was statistically significantly lower in IS-III than IS-II under axial loading (*p* < 0.001), whereas there was no significant difference between the two models in the 15° and 45° loading directions (*p* = 0.088, *p* = 0.408, respectively), as shown in Table 3.

IS-III has a smaller cross-sectional area of thread than IS-II, but IS-III showed a higher bone-forming volume than IS-II. Additionally, it was confirmed that IS-III had a higher volume than IS-II at all measurement points, as shown in Table 4.

For various loading directions, the stress distributions of the single surgical models corresponding to the two different fixtures are shown in Figure 7.

The PVMS of the implant systems and cement are indicated in Table 5. The fixture PVMS values were higher in IS-III than IS-II under all loading directions. On the other hand, the PVMS in the IS-II were higher than the IS-III in both abutment screw and abutment. In particular, the stress of the abutment was shown as approximately 70 MPa higher in IS-II than IS-III. In the cement area, there was no difference between the two fixture models under any loading direction.

In the mechanical test, maximum loads were measured as 1050.01 ± 44.37 N and 1085.78 ± 45.74 N in IS-II and IS-III, respectively, as shown in Figure 8. However, the maximum load values were not significantly different between the two different implant systems (*p* = 0.3068).

## 4. Discussion

Dental implants consist of a combination of various design factors, including fixture connection type, fixture shape, size, abutment type, thread shape and pitch, and screw type. Although research on each design factor has been performed in previous literature [22,23,25,26,27], and the performance of dental implant systems has previously been assessed by mechanical testing using a universal testing machine [47], when multiple design changes are made, the results derived by previous research do not apply. In order to analyze the differences resulting from design changes quickly and to derive trends of different implant systems, the FEA technique has mainly been carried out. It was used for the first time in dentistry for structural analysis of the bone and implant systems in 1976 [48]. Subsequently, the FEA technique has been widely used to evaluate and predict the biomechanical characteristics and tendency for design changes in dental implant systems [48].

Therefore, this work was carried out using FEA to compare the structural stability and biomechanical characteristics relative to design changes, including changes to the thread pitch and thickness of the fixture, the cutting edge, and the thickness and shape of the collar area, between IS-II and IS-III. Additionally, we compared the PVMS values of the implants and cement to predict the tendency to implant fixation failure. To compare the performance of the two dental implants, mechanical testing was performed to derive the maximum load using a universal testing machine.

In this study, two loading conditions were applied to the dental implant procedure in a similar manner to a real treatment situation [28,35]. Firstly, a preload was applied to complete tightening of the contact surface between the outer surface of the abutment and the inner surface of the fixture. Then, masticatory force was applied at the first molar under various loading directions. This force condition assumed the oral environment that the first molar in the mandible bone was laterally biased towards bolus in posterior occlusion closure [36,49]. In addition, the axial load was applied to six cups on the molar surface and the oblique load was applied to three fossae and three cups to account for the various loads on the teeth [36]. In the stress distribution of the abutment screw, highest stress was identified on the fourth thread position under all loading directions, as shown in Figure 9e. In particular, the stress was only higher on the fourth screw thread position under an axial loading direction. However, Zhang [35] reported that stress concentration was shown in the screw thread of the neck area of the abutment under an oblique loading direction, and our study also identified that the highest stress in the oblique loading direction was increased at the neck position of the screw. The tendencies of stress for the abutment screw were similar between the two models. Based on the results mentioned above, fatigue failure was predicted at the fourth thread position and the neck of the abutment screw. Since IS-III has a more desirable stress distribution than IS-II, fatigue fracture of the abutment screw was expected to occur in IS-II.

In the cortical bone, the mean values of Max EQS were lower in IS-III than IS-II for all loading directions. Since the lower Max EQS was measured for the IS-III, it was shown that IS-III has lower stress in the surface of the cortical bone around the fixture. Thus, the cortical bone with the IS-III has less damage and a consequently higher success rate [42]. Similar to the previous literature [26,27,35,42], our results showed that the stress mainly is concentrated in the cortical bone area. Whereas the Max EQS on the cancellous bone was statistically higher in IS-III than IS-II for all loading directions, IS-III showed a lower stress distribution for the cortical bone but higher stress distribution for the cancellous bone than IS-II. Hence, the stress in the cancellous bone was more desirably transferred in the IS-III model, which may disperse the concentrated stress in the cortical bone, as shown in Figure 9a,b. Additionally, the above results are considered to show that the primary stability in IS-III may be superior to IS-II.

The Max EQS of the nerve around the cancellous bone was significantly lower in IS-III than IS-II at axial load, however the IS-II and IS-III were not significantly different at oblique loading directions. Similar to the above result, the minimum principal strain was statistically lower in IS-III than IS-II at axial load, and there were no significant differences at oblique loading directions. These results were related to the volume according to the fixture design. The bone volume of the IS-III model was higher than the IS-II model, which means that more bone can be formed around the fixture. In addition, it indicates that when the same load is applied, bone formation is increased and the stress transfer from the tooth to the nerve is dispersed to decrease the stress concentration and deformation in the nerve. These results suggest that IS-III may reduce the possibility of pain caused with stress in the nerve tissue under axial load, as shown in Figure 9f.

To compare the structural and biomechanical stability of the implants and cement, the PVMS values of each part were calculated. In the stress distribution of the assembled model, the PVMS was generated at the point where the hex component of the abutment and the inner surface of the fixture were in contact under axial and 45° loading directions, as shown in Figure 7a,b. On the other hand, the PVMS occurred in the fixture near the cortical bone under a 15° loading direction, as shown in Figure 7a,b. In the abutment and abutment screw under all loading directions, IS-II showed higher PVMS than IS-III, as shown in Figure 9d,e, whereas the stress of the fixture was lower in IS-II than IS-III, as shown in Figure 9c. It was reported in previous literature that the loading conditions and implant design affect the stress distribution in the bone and implants [50]. In this study, we also identified that the stress distribution differs according to the fixture design and the various loadings.

Moreover, to compare the dental implant systems, the maximum loads for the two fixture designs were derived. The results showed that the IS-III has the tendency of a higher maximum load than IS-II, but the difference was not significant. However, in the FEA results, we confirmed that the high stresses at the platform of IS-II were distributed more widely than IS-III, as shown in Figure 9c. This result means that the thickness reduction induced the decrease of structural stability.

As previous literature reported [51,52,53], the results of investigations into the relationship between aging and alveolar bone reduction reveal a sharp increase in bone reduction from 30 through to 50 years of age, with a maintaining of the bone reduction rate in years 60 and older. Therefore, when a dental implant is placed in an older patient with reduced alveolar bone, IS-III is considered to be superior to IS-II active because it generates less stress and deformation in the nerve. 

In this study, FEA and mechanical testing were performed to compare the biomechanical and primary stability between IS-II and IS-III using a single surgical model and a universal testing machine. This study confirmed the greater biomechanical and primary stability of IS-III compared to IS-II with respect to the thread pitch and shape and the cutting edge.

## 5. Conclusions

On the basis of our results, the optimized design (IS-III) has a more desirable stress distribution in the surrounding bone region. This result can contribute to the favorable stress distribution on the surrounding bone through the implant fixture after loading the implant prosthesis, which can affect the long-term longevity of the implant. In addition, stress of the cortical bone in the IS-III was relatively low compared to IS-II. These findings imply that bone heat generated during implant installation can occur less, and based on this result, IS-III may be more effective in mandibles with a large amount of cortical bone. 

Through this study, the superiority of IS-III was demonstrated using biomechanical evaluations, but further pre-clinical/clinical research and fatigue testing are required for data on aspects such as osteointegration relating to characteristics of the micro-grooves at the platform of the fixture and the fatigue strength. In addition, a recent study has reported an antibacterial internal coating to prevent bone reduction around fixtures of an internal type, and thus, the antibacterial coating of fixture internals should be considered in future implant development [54].

## Figures and Tables

**Figure 1 materials-12-02749-f001:**
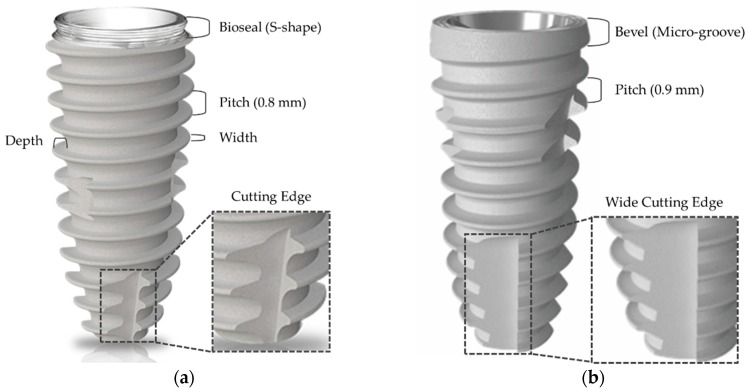
Implant macro design parameters in dental fixtures. (**a**) CMI IS-II active; (**b**) CMI IS-III active.

**Figure 2 materials-12-02749-f002:**
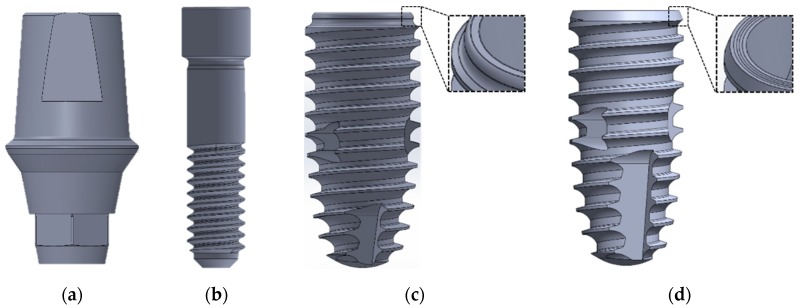
The two different implant fixture models used in the present study. (**a**) Abutment; (**b**) abutment screw; (**c**) CMI IS-II active; (**d**) CMI IS-III active.

**Figure 3 materials-12-02749-f003:**
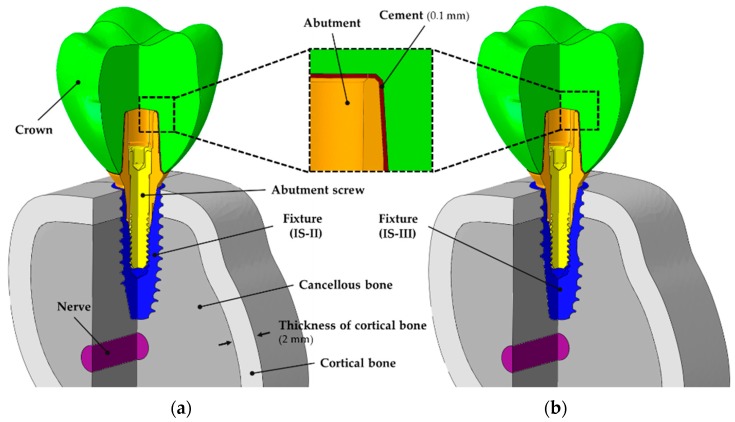
Cross-sectional view of three-dimensional finite element models for a single-tooth surgery showing the internal assembly structure and cement layer. (**a**) CMI IS-II active fixture and (**b**) CMI IS-III active fixture.

**Figure 4 materials-12-02749-f004:**
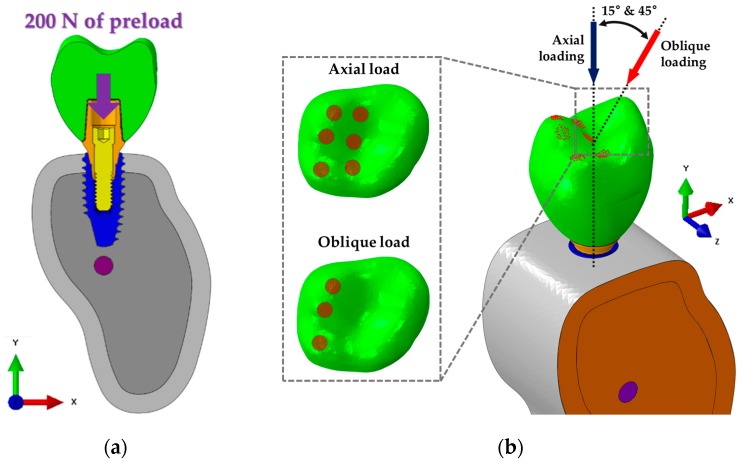
Boundary and loading conditions of FEA models (IS-II and IS-III). Both sides of the bone model (orange) are fixed in all axes. (**a**) Application of 200 N in the axial direction to the screw assuming the complete connection of the fixture and abutment. (**b**) Total load of 100 N applied to 187 nodes on 6 cups in the axial direction of the fixture and total loads of 100 N (15°) and 30 N (45°) applied to 86 nodes on 3 fossae and 3 cups in a non-axial direction considering the various directional loads.

**Figure 5 materials-12-02749-f005:**
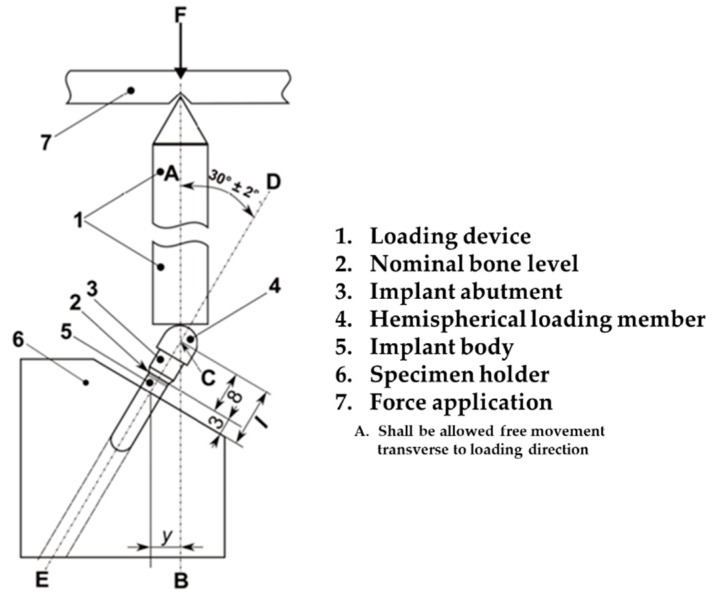
Schematic of the static shear-compression loading experiment. Schematic of the test setup based on the revised ISO 14801:2016.

**Figure 6 materials-12-02749-f006:**
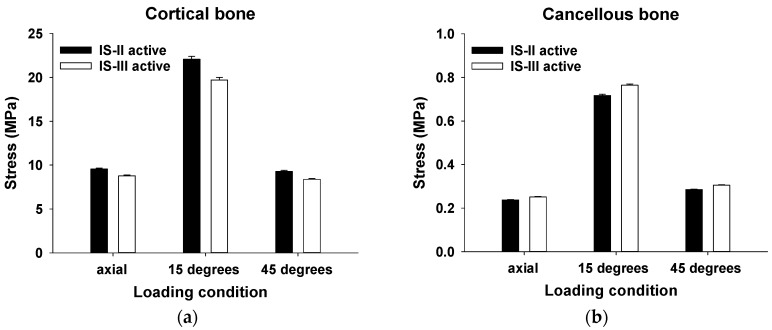
The Max EQS values in (**a**) cortical bone, (**b**) cancellous bone, (**c**) and the nerve and (**d**) the minimum principal strain of the nerve under loading of 100 N (axial), 100 N (15°), and 30 N (45°).

**Figure 7 materials-12-02749-f007:**
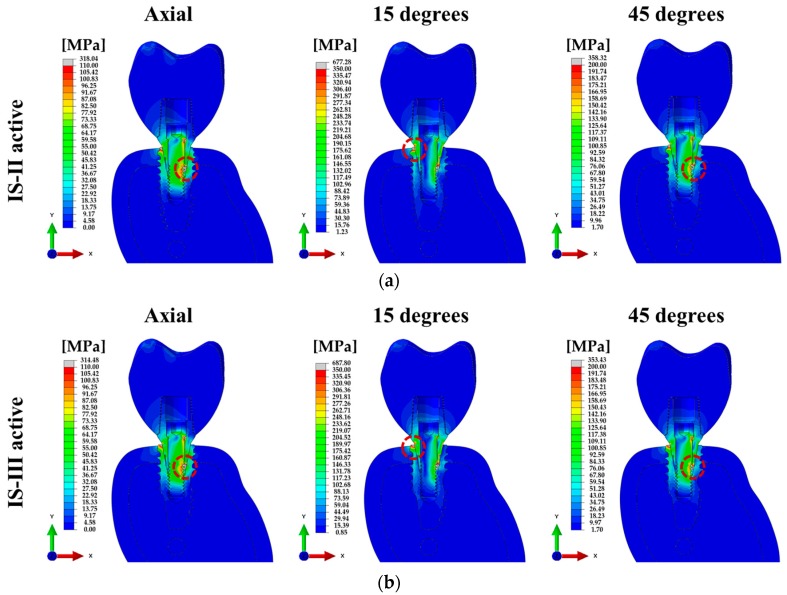
Cross-sectional view of the von Mises stress for single-tooth surgical FE models under various loading directions. (**a**) IS-II and (**b**) IS-III.

**Figure 8 materials-12-02749-f008:**
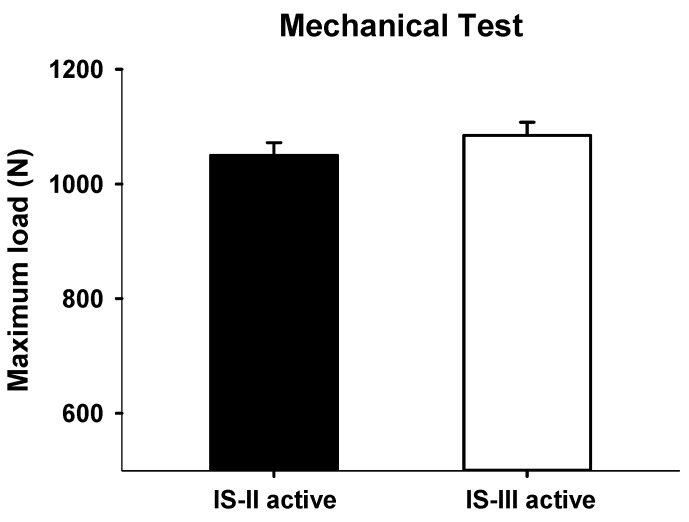
Maximum loads (N) of IS-II and IS-III.

**Figure 9 materials-12-02749-f009:**
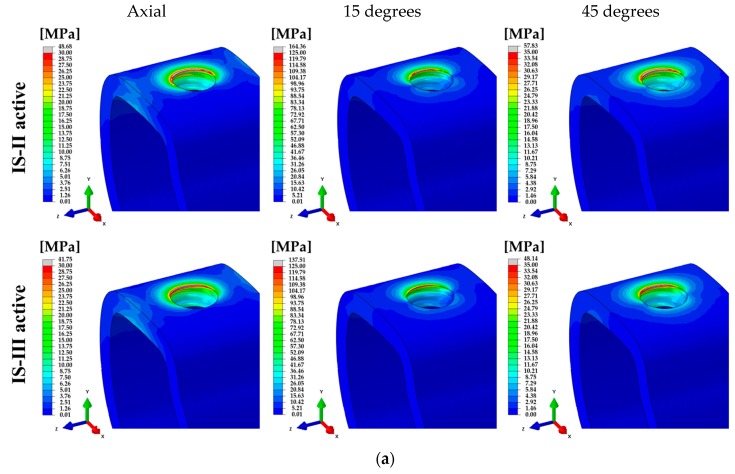
The distribution of von Mises stress under various loads. (**a**) Cortical- and (**b**) cancellous-bones, (**c**) fixture, (**d**) abutment, (**e**) abutment screw, and (**f**) nerve for IS-II and IS-III models.

**Table 1 materials-12-02749-t001:** Material properties applied in the finite element analysis (FEA) models.

Components	Young’s Modulus (MPa)	Poisson’s Ratio
Crown (Zirconia) [32]	205,000	0.19
Abutment (Ti grade 5) [33]	114,000	0.33
Fixture (Ti grade 4) [34]	105,000	0.34
Abutment screw (Ti grade 5) [33]	114,000	0.33
Cortical bone [35]	13,000	0.30
Cancellous bone [35]	690	0.30
Nerve canal [36]	70	0.45
Cement [37]	10,310	0.24

**Table 2 materials-12-02749-t002:** Mesh size and number of elements and nodes of the FEA models.

Components	Elements		Nodes		Mesh Size (mm)	
	IS-II	IS-III	IS-II	IS-III	Maximum	Minimum
Crown	121,740		25,678		0.30	0.15
Abutment	88,637		21,139		0.15	0.05
Fixture	125,966	48,692 (up) 104,432 (down)	24,396	12,300 (up) 23,594 (down)	0.15	0.05
Abutment screw	60,588		13,836		0.15	0.03
Cement	85,425		20,482		0.1	0.03
Cortical bone	287,389	295,533	62,917	64,492	1.00	0.15
Cancellous bone	275,822	273,597	57,108	56,765	1.00	0.15
Nerve	58,241		15,781		0.15	0.05

**Table 3 materials-12-02749-t003:** The Max EQS and minimum principal strain (compressive strain) results (mean ± SEM) of IS-II and IS-II fixtures for cortical bone, cancellous bone, and the nerve surface (* *p*-value < 0.01).

**Components**	**Direction**	**Maximum Equivalent Stress (MPa)**				
		**IS-II**	***n***	**IS-III**	***n***	***p*** **-value**
Cortical bone	100 N (Axial)	9.56 ± 0.10	4586	8.78 ± 0.09	3906	<0.001 *
	100 N (15°)	22.07 ± 0.32		19.70 ± 0.29		<0.001 *
	30 N (45°)	9.28 ± 0.12		8.38 ± 0.10		<0.001 *
Cancellous bone	100 N (Axial)	0.24 ± 0.002	8192	0.25 ± 0.002	7450	<0.001 *
	100 N (15°)	0.72 ± 0.005		0.76 ± 0.005		<0.001 *
	30 N (45°)	0.29 ± 0.002		0.31 ± 0.002		<0.001 *
**Components**	**Direction**	**Maximum Equivalent Stress (kPa)**				
		**IS-II**	***n***	**IS-III**	***n***	***p*** **-value**
Nerve	100 N (Axial)	18.80 ± 0.10	6031	18.30 ± 0.09	6031	<0.001 *
	100 N (15°)	19.80 ± 0.21		19.40 ± 0.20		0.168
	30 N (45°)	8.90 ± 0.07		8.80 ± 0.07		0.775
**Components**	**Direction**	**Minimum Principal Strain (×10^−3^)**				
		**IS-II**	***n***	**IS-III**	***n***	***p-*** **value**
Nerve	100 N (Axial)	2.796 ± 0.015	6031	2.719 ± 0.015	6031	<0.001 *
	100 N (15°)	2.339 ± 0.025		2.278 ± 0.025		0.088
	30 N (45°)	0.759 ± 0.007		0.751 ± 0.006		0.408

**Table 4 materials-12-02749-t004:** The values of overall volume, one pitch volume, and cancellous bone volume at the apex of the fixture for IS-II and IS-III.

Type	Measurement Point (mm^3^)		
	One Pitch	Pitches and Apex	Apex Region
IS-II	2.68	33.42	77.99
IS-III	2.86	35.90	78.05

**Table 5 materials-12-02749-t005:** Peak von Mises stress (PVMS) results of IS-II and IS-III for the abutment, fixture, abutment screw, and cement.

Loading Conditions	PVMS (Mpa)							
	Abutment		Fixture		Abutment Screw		Cement	
	IS-II	IS-III	IS-II	IS-III	IS-II	IS-III	IS-II	IS-III
100 N (Axial)	318.04	314.48	257.16	304.65	289.58	256.66	15.22	15.17
100 N (15°)	650.30	583.47	677.28	687.80	485.04	431.95	43.12	43.18
30 N (45°)	358.32	353.43	229.58	270.15	290.52	286.19	15.89	15.86

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
