# Peer review of "Optimized Dental Implant Fixture Design for the Desirable Stress Distribution in the Surrounding Bone Region: A Biomechanical Analysis"

_materials, 2019, doi:10.3390/ma12172749_

Round 1

Reviewer 1 Report

Comments to the Authors:

The objective of this study was to compare the initial stability of two different implant systems using a 3‐dimensional (3D) mandible bone model with finite element analysis (FEA) and mechanical testing for maximum loads.  Some suggestions are listed as following:

Table 1 is the material properties used in the finite element models. The information seems copy from reference of 36. Lee, H.J.; Park, S.Y.; Noh, G.W. Biomechanical analysis of 4 types of short dental implants in a resorbed mandible. J. Prosthet. Dent. 2019, 121, 659‐670.  However, the cited references of 32-37 are all different from original article. In Table 3, the (MPa) should move to Maximum Equivalent Stress rather than Cortical bone. In Line 235, what is the unit for maximum loads of 1050.01 ± 44.37 and 1085.78 ± 45.74? In Table 3, the Maximum Equivalent Stresses in Cancellous bone for IS-II and IS-III are 0.24±002 and 0.25±0.002, respectively. It was claimed the p < 0.001.  Please confirm. In Table 3, the Maximum Equivalent Stresses in nerve for IS-II and IS-III are 18.8±10 and 18.9±0.009, respectively. It was claimed the p < 0.001.  Please confirm. Under finite element analysis (FEA), the stress distribution of the cortical bone was statistically significantly higher in IS‐II than IS- III shown in Table 3.   In the mechanical test, maximum loads in IS‐II and IS‐III, indicating the maximum load values were not significantly different.  The trends are difference between the FEA simulation and measurement.  Then, how we make a fair judgement?

Author Response

Dear Reviewer

Reviewer 2 Report

Dear authors,

I have read your manuscript with great interest and I think it could sound very interesting for the readers.

Nonetheless, before admitting this manuscript for publication I suggest you to make some changes and then resubmit it for subsequent evaluation.

Major changes:

In introduction (Line 42) you write about "esthetic dissatisfaction of the patient". Are you referring to the situation in which the tooth to be replaced by the implant is in an aesthetic zone? In that case I would replace the sentence and write about the increasing of costs for the temporary prosthesis. Otherwise there is no need for aesthetic replacement during wound healing and osseointegration in posterior regions. In discussion (Line 259) you write "occlusal force was applied at the first molar". I think you refer to the computational simulation of strenght that resembles the one of the first molar during mastication. In that case please rewrite in a more correct manner adding also this reference (Piancino MG et al. From periodontal mechanoreceptors to chewing motor control: A systematic review. Archives of Oral Biology, 2017, 78, 109-121). Discussion (Paragraph line 307-311): you express the concept of relationship between oral alveolar bone loss and aging that is a fundamental point to be underlined writing on a topic like the one of your manuscript. Please add references regarding bacterial role in the generation of periodontal and perimplant disease and its correlation with systemic diseases (Mohammed H et al, Oral dysbiosis in pancreatic cancer and liver cirrhosis: A review of the literature Biomedicines, 2018, 6, 115; Patini R et al., Correlation between metabolic syndrome, periodontitis and reactive oxygen species production. A pilot study, Open Dent J, 2017, 11, 621-627) Conclusion (Last paragraph line 324-327): please insert in conclusions a reference to new strategies in avoiding bacterial bone resorption and lack of implant osseointegration.

Minor changes:

In results (Line 228): please explicit such acronym: PVMS.

Regards

Author Response

Dear Reviewer

Round 2

Reviewer 2 Report

Dear authors,

thanks for having satisfactorily addressed all my comments.

Best regards